# New Insights from Locally Resolved Hydrodynamics in Stirred Cell Culture Reactors

**Fabian Freiberger [1], Jens Budde [1], Eda Ateş [1], Michael Schlüter [2] , Ralf Pörtner [1,\*] and Johannes Möller [1]**

[1] Institute of Bioprocess and Biosystems Engineering, Hamburg University of Technology, Denickestr. 15, 21073 Hamburg, Germany; fabian.freiberger@tuhh.de (F.F.); jens.budde@tuhh.de (J.B.); eda.ates@tuhh.de (E.A.); johannes.moeller@tuhh.de (J.M.)

[2] Institute of Multiphase Flows, Hamburg University of Technology, Eißendorfer Str. 38, 21073 Hamburg, Germany; michael.schlueter@tuhh.de

\* Correspondence: poertner@tuhh.de; Tel.: +49-40-42878-2886

**Abstract:** The link between hydrodynamics and biological process behavior of antibody-producing mammalian cell cultures is still not fully understood. Common methods to describe dependencies refer mostly to averaged hydrodynamic parameters obtained for individual cultivation systems. In this study, cellular effects and locally resolved hydrodynamics were investigated for impellers with different spatial hydrodynamics. Therefore, the hydrodynamics, mainly flow velocity, shear rate and power input, in a single- and a three-impeller bioreactor setup were analyzed by means of CFD simulations, and cultivation experiments with antibody-producing Chinese hamster ovary (CHO) cells were performed at various agitation rates in both reactor setups. Within the three-impeller bioreactor setup, cells could be cultivated successfully at much higher agitation rates as in the single-impeller bioreactor, probably due to a more uniform flow pattern. It could be shown that this different behavior cannot be linked to parameters commonly used to describe shear effects on cells such as the mean energy dissipation rate or the Kolmogorov length scale, even if this concept is extended by locally resolved hydrodynamic parameters. Alternatively, the hydrodynamic heterogeneity was statistically quantified by means of variance coefficients of the hydrodynamic parameters fluid velocity, shear rate, and energy dissipation rate. The calculated variance coefficients of all hydrodynamic parameters were higher in the setup with three impellers than in the single impeller setup, which might explain the rather stable process behavior in multiple impeller systems due to the reduced hydrodynamic heterogeneity. Such comprehensive insights lead to a deeper understanding of the bioprocess.

**Keywords:** CHO DP-12; computational fluid dynamics; bioreactor characterization; hydrodynamic gradients; process development; critical shear stress; Kolmogorov length scale; operational space



## 1. Introduction

Mammalian cell culture processes are state-of-the-art for the production of therapeutic antibodies. However, the influence of the bioreactor hydrodynamics on the cell culture process are still not fully understood [1–6]. Therefore, bioreactor design and scale-up in today's biopharma industry rely mostly on empirical correlations, experience, and engineering heuristics. Common methods for scaling-up mammalian cell-based production processes aim to keep the reactor geometry and certain process parameters such as the average volumetric power input constant [7–11].

These "rules of thumb" methods are limited as they can hardly be used to describe or estimate an appropriate operation range with respect to shear effects on cellular behavior. Platas et al. [12] observed an approx. constant cell specific growth rate, $\mu_{max}$, over a broader stirrer operation range in reactor systems with multiple impellers compared to reactor systems with only one impeller. The cell growth rate in reactors with multiple impellers decreased much slower with increasing agitation rates and corresponding mean

power inputs than in reactors with less impellers. Therefore, shear effects seem to be less pronounced in the case of multiple impellers.

An approach frequently discussed in the literature to predict critical conditions with respect to shear effects to be expected in stirred tank bioreactors is the Kolmogorov eddy length scale [12,13], for which cell harm is predicted if the eddies are in the order of magnitude as the cell diameter. This approach should predict an acceptable power input—and thus an adequate agitation rate—to be estimated without extensive experiments. Conventionally, the system averaged energy dissipation rate is used to calculate the Kolmogorov length scale, which does not consider local gradients, even if the local power input can differ by several orders of magnitude within the reactor. Furthermore, this length scale is usually compared considering an average cell diameter, even though the cell diameter is widely distributed. However, while this hypothesis seems to be proven for microcarrier cultures, it is still under discussion for suspension cells [14–19]. A broad variety of studies on lethal and sub-lethal responses of mammalian cell cultures to shear stress has been published where a detailed overview can be found in Chalmers et al. The investigated magnitude of the average volumetric power input ranges from $10^1$ to almost $10^9$ W m$^{-3}$ [3]. However, the vast majority of these studies only considered the estimated or averaged hydrodynamics.

Taken together, more insights into locally resolved hydrodynamics are needed. Since local maxima can reach much higher values than the averaged, locally resolved hydrodynamics could provide more detailed information on the hydrodynamics. These can be identified and quantified by the use of computational fluid dynamics (CFD) methods, which have become an established tool for cell culture process development [5,10,20–22]. CFD can be used to predict the locally resolved hydrodynamic behavior of cell cultivation systems and has already been successfully applied for the design of stem cell expansion processes in stirred tank reactors and wave bags with microcarriers [23–25].

In this study, we investigated to what extent the cellular and hydrodynamic effects change with different spatial hydrodynamics of different stirrers. Furthermore, it was discussed whether locally resolved hydrodynamics could help to explain the cellular effects. Therefore, the hydrodynamics, mainly flow velocity, shear rate, and power input, in a single- and a three-impeller bioreactor setup were analyzed by means of CFD simulations. Then, cultivation experiments with antibody-producing Chinese hamster ovary (CHO) cells were performed at various agitation rates in both reactor setups. Finally, conventional process design parameters such as the average volumetric power input and the Kolmogorov length scale were evaluated by means of the obtained data. As the model system, a reactor setup was chosen that could be equipped with various numbers of impellers and operated without baffles at constant culture volumes. Because surface aeration is sufficient for a wide range of operation conditions, the impact of bubble aeration was neglected in these investigations.

## 2. Materials and Methods

### 2.1. Cultivation Procedures

The cell line used in all cultivation experiments was CHO DP-12, which was kindly provided by Prof. Dr. Thomas Noll (University of Bielefeld, Bielefeld, Germany). All cultivation experiments were performed in TC-42 medium (Xell AG, Bielefeld, Germany), supplemented with 8 mmol L$^{-1}$ L-glutamine and 200 nmol L$^{-1}$ methotrexate.

#### 2.1.1. Pre-Culture

Pre-cultivations were carried out in 125 mL shake flasks with 40 mL culture volume for the first culture after thawing and 250 mL shake flasks with 80 mL culture volume for expanding the cells [26]. Culture conditions were adjusted to 37 °C, 5% $CO_2$, 85% relative humidity, and 200 rpm. From pre-cultures, $15 \times 10^7$ cells in total were harvested, centrifuged for 10 min at $300 \times g$, and resuspended in 10 mL fresh medium for inoculating the stirred tank bioreactor with an initial cell concentration of $1 \times 10^6$ cells mL$^{-1}$.

### 2.1.2. Main Culture

For bioreactor cultivations, the stirred tank bioreactor Vario 1000 (MDX Biotech GmbH, Nörten-Hardenberg, Germany) was used with 150 mL culture volume and headspace aeration by gas mixtures of air and $CO_2$. Only at the end of the cultivations at cell densities of around $10 \times 10^6$ cells $mL^{-1}$, the reactor was additionally bubble-aerated with pure oxygen. Thus, effects of gas bubbles and additional turbulences, caused by baffles could be neglected and observed effects could be referred to the impellers. The dissolved oxygen tension (DO) was controlled at 40% of air saturation in all cultivations. pH was controlled at 7.1 by the addition of $CO_2$ via headspace or by adding 0.5 M $Na_2CO_3$. The temperature was set to 37 °C. To investigate the effect of hydrodynamics caused by stirring on the cell culture process, cultivation experiments were performed at different agitation rates for a reactor setup with one pitched blade impeller (MDX Biotech GmbH, Nörten-Hardenberg, Germany) (single impeller system, SIS) and a setup with two additional six blade impellers (self-made at Hamburg University of Technology, Nörten-Hardenberg, Germany) (triple impeller system, TIS) (see Appendix A Figure A1). The agitation rates chosen for the SIS were 200 to 1400 rpm in 200 rpm steps, while TIS cultivations at agitation rates of 770, 930, 1080, 1200 and 1400 rpm were run. In all experiments, the pitched blade impeller was operated in the down flow direction. Agitation rates for the TIS were chosen while following a discarded working hypothesis.

### 2.2. Analytical Methods

Cell densities were determined with the Z2 Particle Counter (Beckman Coulter, Brea, CA, USA). All particle distributions were recorded in triplicate. The cell viability was measured using the flow cytometer CytoFLEX (Beckman Coulter, Brea, CA, USA) after staining the cells with 1 µg $mL^{-1}$ DAPI. Samples above $2 \times 10^6$ cells $mL^{-1}$ were diluted ten-fold before staining. Antibody concentrations were determined with Protein-A binding sensors in the OCTET according to the manufacturer's protocol (Pall Corporation, Port Washington, NY, USA). For a comparison of the processes, in addition to growth curves, the space time yield (STY) related to the antibody concentration was calculated as the quotient of the maximum antibody concentration and the cultivation time until the maximum concentration was reached.

### 2.3. CFD Simulations

All CFD models were developed in COMSOL Multiphysics 5.3 to 5.5 (COMSOL AB, Stockholm, Sweden). The reactor geometry was fully implemented in COMSOL, except for the six blade impeller, which was imported from a CAD file. Reactor vessel and impellers were modeled as accurately as possible and necessary. Additionally, all reactor inserts, namely the impeller shaft, temperature, pH and DO probes as well as sampling tubes and aeration tubes were implemented. As the physics module, the mixture module was chosen with turbulent flow conditions. The module contains the Reynolds Averaged Navier–Stokes equation and continuity equation as governing equations. Standard parameters were used to model the turbulence with the k-ε-model. The geometry was meshed to 1.69 million mesh elements for the SIS and 1.78 million mesh elements for the TIS. To solve the stationary studies, the PARDISO solver was set up as a direct block structured solver. For further, detailed information about the CFD models, see the model reports attached as Supplementary Materials (Files S1 and S2). Due to numerical reasons, agitation rates of up to 800 rpm for the reactor setup with a single impeller and 600 rpm for the reactor setup with three impellers were simulated. Velocities and shear rates above the given agitation rates were extrapolated linearly while the energy dissipation rates were extrapolated quadratically, according to the simplified dependency given by Hu et al. [1]. The computed values from the CFD simulations and extrapolations can be found in Appendix A (Tables A1 and A2). A mesh refinement study was performed for all shown models (not shown). No dependencies on the degree of meshing were observed.

To highlight local distributions of hydrodynamic parameters, the computed velocities, shear rates, and energy dissipation rates were averaged over horizontal cut planes along the reactor height in steps of one millimeter. The obtained mean values were then plotted in comparison to the reactor height.

### 2.4. Calculation of the Critical Energy Dissipation Rate

The Kolmogorov length scale represents the size of the smallest turbulence eddies in a fluid flow and can be calculated from the energy dissipation rate $\varepsilon$, the kinematic viscosity $\nu$, and the density $\rho$ of the fluid.

$$\lambda = \left( \frac{\nu^3 \rho}{\varepsilon} \right)^{\frac{1}{4}} \tag{1}$$

According to the hypothesis, significant cell damage occurs when the smallest turbulence eddies are within the same order of magnitude as the cell diameter $d_c$. Thus, the equation for the Kolmogorov length was solved for the critical energy dissipation rate $\varepsilon_{krit}$ where the cell diameter $d_c$ is the cell diameter, which is in the same order as the Kolmogorov length $\lambda$, $\nu$ is the kinematic viscosity, and $\rho$ is the density of the fluid.

$$\varepsilon_{krit} = \frac{\nu^3 \rho}{d_c^4} \tag{2}$$

Distributions of the cell diameter $d_c$ were determined alongside the cell number in the Z2 Particle Counter. More details on the relation of turbulence eddies and cell damage according to the Kolmogorov length scale hypothesis can be found in Section 3.3.

## 3. Results

In the following, the results from the CFD study as well as from the cell cultivation experiments are shown for the single-impeller setup (SIS) and the three-impeller setup (TIS). Cultivation results are presented for both reactor setups to demonstrate the biological responses to different hydrodynamic conditions in the different setups. Finally, it was examined to what extent the observed effects can be explained on the basis of the hydrodynamic parameters determined via CFD simulations, where mainly the fluid velocity, power input, or energy dissipation rate were considered. It was investigated whether locally resolved parameters and parameter distributions had an advantage in this respect compared to the averaged values. This is further discussed in Section 4.

### 3.1. Characterization of Hydrodynamics of Single- and Multiple-Impeller Setups with CFD Methods

For both bioreactor setups (SIS and TIS, respectively), the hydrodynamic parameters fluid velocity u, shear rate $\gamma$, and energy dissipation rate $\varepsilon$ were evaluated in the CFD study. Locally resolved values of these parameters were determined along the reactor height.

For example, the hydrodynamic parameters u and $\varepsilon$ for 400 rpm are shown in Figures 1 and 2a (SIS) and Figure 2b (TIS), plotted in longitudinal section (for shear rate $\gamma$, see Figure A2). To gain a deeper numerical insight into locally resolved values, the CFD computed values for velocity, shear rate, and energy dissipation rate were averaged for horizontal slices along the reactor height. The mean values were then plotted over these slices to quantitatively visualize locally resolved values for all simulated agitation rates (slice plots in Figures 1 and 2c,d).

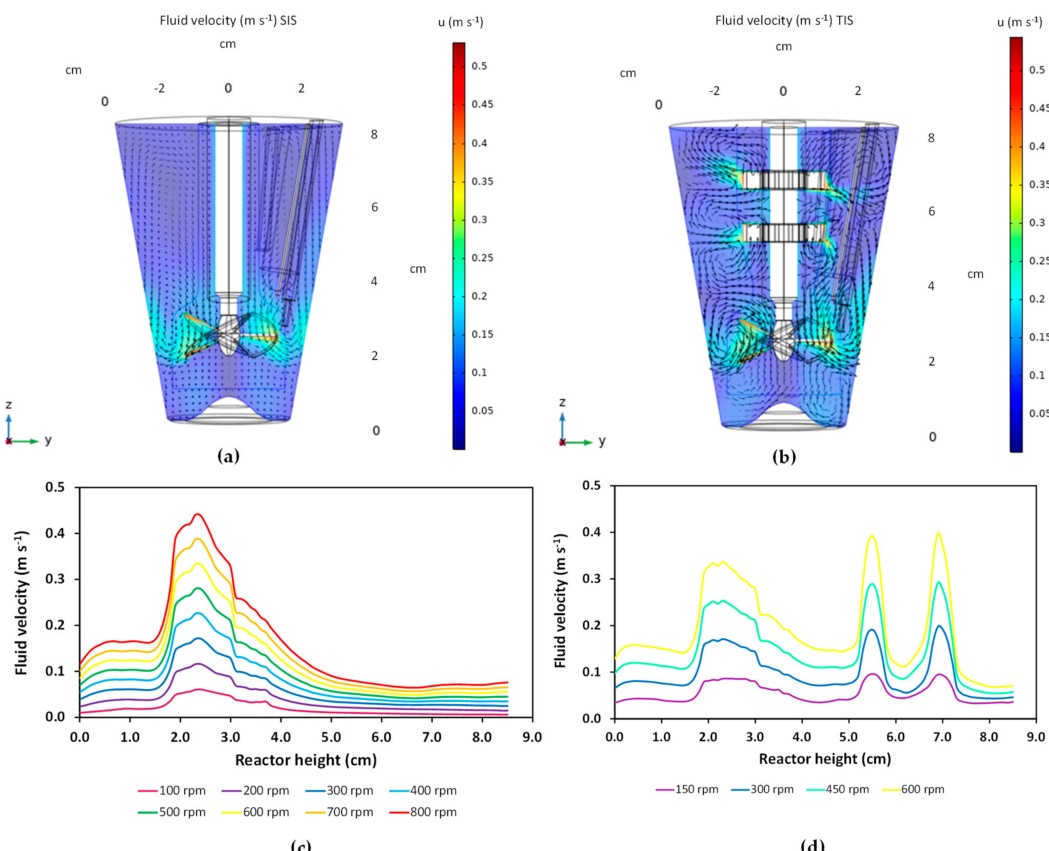

**Figure 1.** Fluid velocity u in the SIS and TIS. (**a**) Fluid velocity in the SIS; (**b**) Fluid velocity in the TIS. The shown data correspond to an agitation rate of 400 rpm. (**c**) Mean fluid velocity along the reactor height (slice plot) in the SIS for different agitation rates; (**d**) mean fluid velocity along the reactor height (slice plot) in the TIS for different agitation rates. In (**c**,**d**), the flow velocity was averaged across the diameter in steps of one millimeter along the reactor height.

From the plots of the fluid velocity u (see Figure 1a,b), the flow pattern in the respective reactor setups can be derived. Both setups were simulated and operated in down flow mode. In the SIS (Figure 1a), the fluid is pressed downward and sideways by the impeller at an approximately 2.5 cm reactor height. Then, it rises to the surface at the vessel walls and comes down again close to the stirrer shaft, yielding a typical axial flow pattern. The fluid velocity u reaches maximal values near the impeller blades up to 0.55 m s$^{-1}$. With increasing distance from the impeller, the fluid velocity u decreases to approx. 0.05 m s$^{-1}$. In the TIS (Figure 1b), the fluid in the upper parts of the reactor is pushed to the side by the six blade impellers, known as radial flow pattern at approx. 5.5 cm and 7 cm reactor height. At the vessel walls, it flows upward and back downward to the impeller close to the stirrer shaft, resulting in a typical radial flow pattern. The system averaged fluid velocity $u_{av}$ was about twice as high as in the SIS with averaged 0.11 m s$^{-1}$.

The plots in Figure 1c,d show general high local values as well as high maxima in narrow areas close to the impeller at about a 2.5 cm reactor height, while large parts of the reactor stayed at values up to five-fold lower. In the TIS, additional peaks could be observed at 5.5 cm and 7 cm reactor height due to the added impellers. It can be seen that the velocity in both setups reached the highest values in regions close to the impellers. Nevertheless, due to the additional impellers, the velocity profile in the TIS was more homogenized than in the SIS.

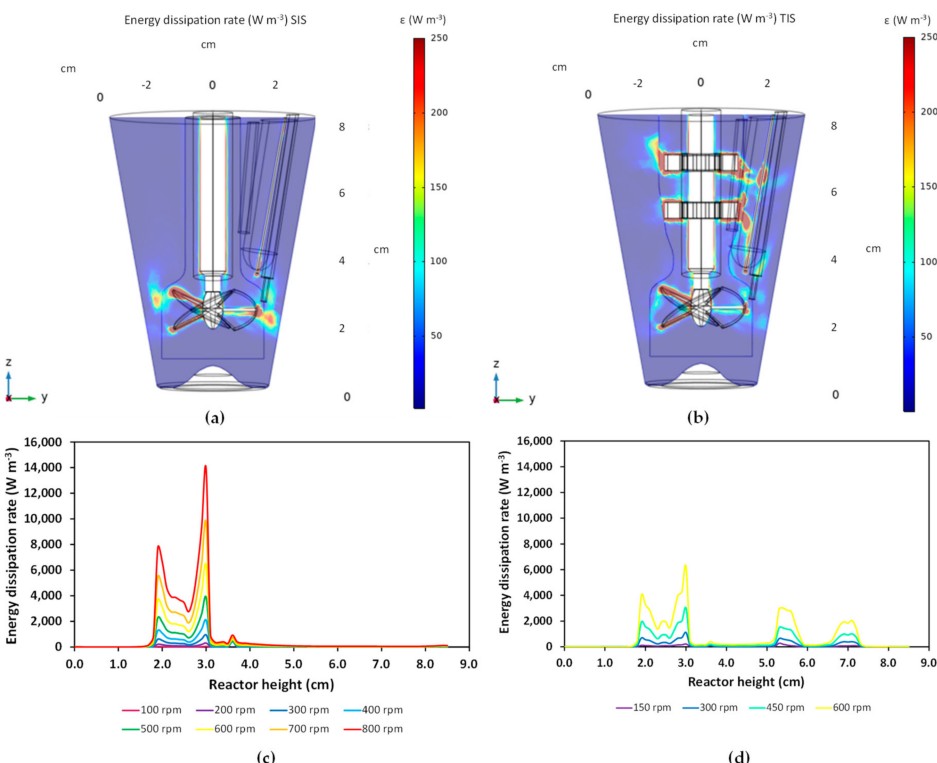

**Figure 2.** Energy dissipation rate $\varepsilon$ in the SIS and TIS: (**a**) energy dissipation rate in the SIS; (**b**) energy dissipation rate in the TIS. The shown data correspond to an agitation rate of 400 rpm. (**d**) Mean energy dissipation rate along the reactor height (slice plot) in the SIS for different agitation rates; (**d**) mean energy dissipation rate along the reactor height (slice plot) in the TIS for different agitation rates. In (**c**,**d**), the energy dissipation rate was averaged across the diameter in steps of one millimeter along the reactor height.

The local energy dissipation rates are shown in Figure 2. In the SIS, the energy dissipation rate $\varepsilon_{max}$ reached values up to 139 kW m$^{-3}$ directly at the impeller (see Figure 2a). The slice plots (see Figure 2c,d) revealed that locally resolved values for the energy dissipation rate showed quite sharp and narrow peaks (also compare Figure A2 for the shear rate $\gamma$). At 800 rpm, the peaks of the energy dissipation rate close to the impeller in 2 cm to 3 cm reactor height are up to 280 times higher compared to the rest of the cultivation system. In this area, the mean energy dissipation rate $\varepsilon$ across the diameter showed a maximum $\varepsilon_{max,SP}$, with extent values of over 14 kW m$^{-3}$, despite it staying between 50 and 100 W m$^{-3}$ in the largest parts of the reactor. The system average volumetric energy dissipation rate $\varepsilon_{av}$, which corresponds to the averaged volumetric power input P/V, did not exceed 110 W m$^{-3}$ for 800 rpm. In the TIS, additional peaks could again be found in the height of the six blade impellers at 5.5 cm and 7.0 cm reactor height. At 600 rpm, the slice plot maximum $\varepsilon_{max,SP}$ of the pitched blade impeller at 3 cm reactor height was over 6 kW m$^{-3}$, and the maximum of the six blade impellers was only slightly over 3 kW m$^{-3}$.

The slice plots (averaged values across the diameter) were obtained for different agitation rates to evaluate the spatial distribution of the investigated hydrodynamic parameters within both reactor systems (see Figures 1 and 2c,d). They all showed peaks for all parameters in the area of the impellers, which were magnitudes higher than in the remaining parts of the system. From the plots, it can be concluded that the characteristic profiles of the six blade impellers were added to the profile of the pitched blade impeller. Pre-investigations to this study showed that these profiles were moved but not altered with a change in the impeller position. In the TIS, maxima from slice plots $\varepsilon_{max,SP}$ for the pitched blade impeller tended to have the same magnitudes as in the SIS for corresponding agitation rates. However, the characteristic peaks of the six blade impellers were added to the peak

of the pitched blade impeller, which led to a higher energy dissipation rate $\varepsilon_{av}$ averaged for the whole system, but lower peaks for the same system averaged energy dissipation rate $\varepsilon_{av}$. Hence, setups with multiple impellers yielded flatter energy dissipation rate $\varepsilon$ distributions for the same averaged power inputs P/V in the same reactor vessels. Further values for agitation rates, averaged dissipation rates $\varepsilon_{av}$, and maximum energy dissipation rates $\varepsilon_{max}$ for both systems can be found in Tables A1 and A2 in Appendix A.

### 3.2. Cultivation Results

Cultivation experiments with CHO DP-12 cells were run in both reactor systems at different agitation rates. For all samples of the cultivation experiments, cell densities and antibody concentrations were quantified. Mean exemplary data containing viable cell densities and viabilities for cultivations at 400, 800, and 1200 rpm in the SIS are shown in Figure 3.

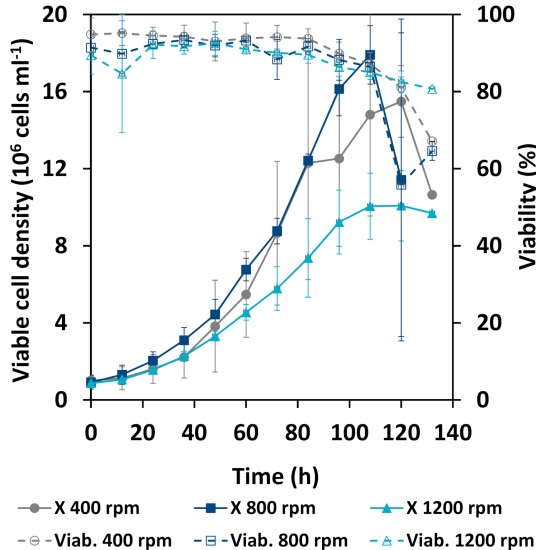

**Figure 3.** Mean growth curves (viable cell density X and viability) for the cultivations in the SIS at different agitation rates and respective averaged power inputs: 400 rpm (16 W m$^{-3}$); 800 rpm (110 W m$^{-3}$); 1200 rpm (356 W m$^{-3}$). Data were averaged from three cultivations each.

Figures 4 and 5 show the results of the cultivation experiments performed in the TIS. Surprisingly, the effect of the agitation rate on maximum cell density, viability, or growth rate seemed to be less pronounced as that for the SIS, even at 1400 rpm. Obviously, the stable operational range was broader when using multiple impellers.

From Figures 3 and 5 (The maximum cell densities are summarized in Figure 5), it can be concluded that the maximal cell density and the cell specific growth rate (not shown) were more or less unaffected by the agitation rate up to 800 rpm. For 1000 rpm and higher agitation rates, a decrease in maximum cell densities and cell specific growth rate was observed. Furthermore, the viability had already started to decrease in processes at high agitation rates above 1000 rpm before reaching the maximum cell density.

With respect to the antibody production behavior of CHO DP-12 cells, the space time yield (STY) was calculated for all performed processes (Figure 6). For the SIS experiments, the highest STYs with up to 2.9 mg L$^{-1}$ h$^{-1}$ are observed between agitation rates of 400 rpm and 800 rpm; for higher agitation rates, the STY declined and at 1400 rpm, the SIS of the antibody concentration in all samples was even below the detection limit of 0.15 g L$^{-1}$. For the TIS, the STYs were approx. constant at around 1.5 mg L$^{-1}$ h$^{-1}$ for agitation rates between 770 rpm and 1400 rpm. These results also support the idea of a broader stable operational range when using multiple impellers.

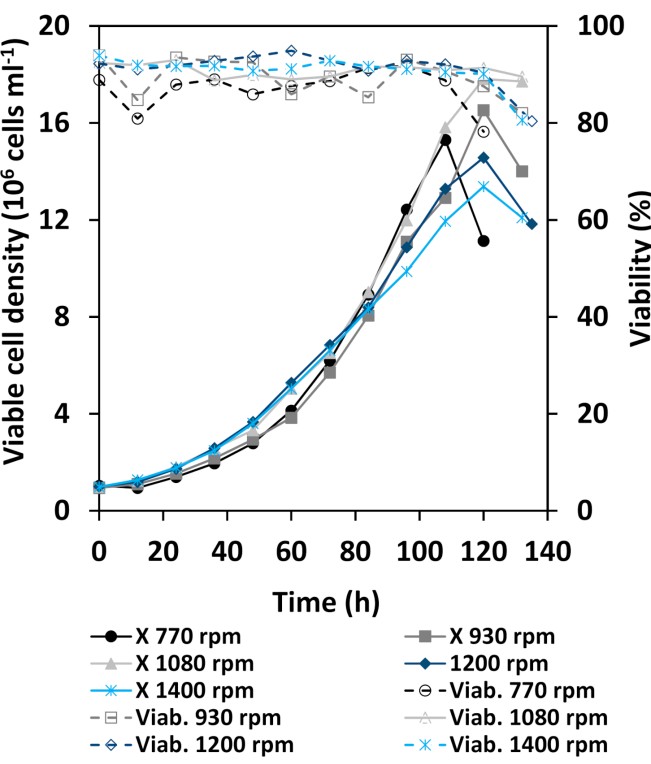

**Figure 4.** Growth curves (viable cell density X and viability) for the cultivations in the TIS at different agitation rates and respective averaged power inputs. The agitation rate settings range from 770 rpm or 454 W m$^{-3}$ to 1400 rpm or 4742 W m$^{-3}$. One experiment was performed at each agitation rate.

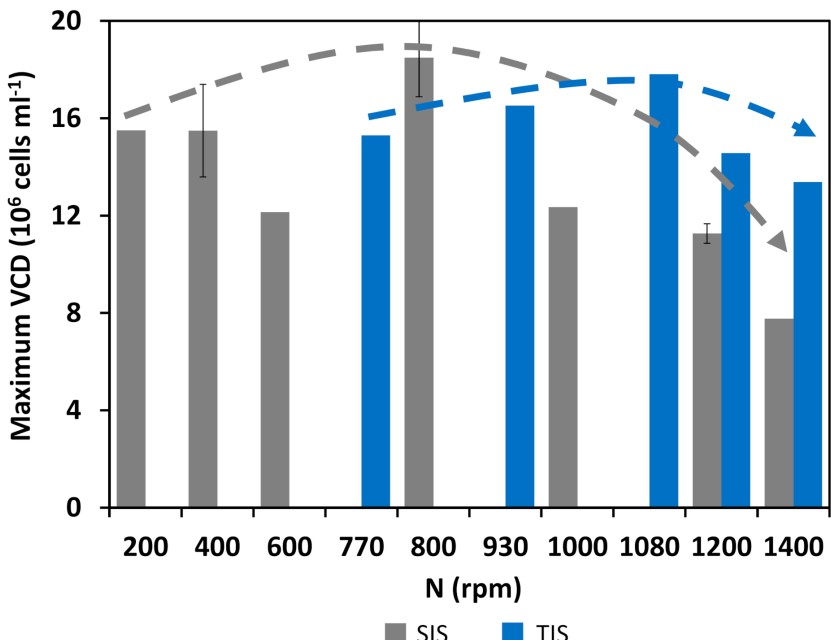

**Figure 5.** Maximum viable cell densities (VCD) compared for experiments in both reactor setups: cultivation results from the SIS (grey bars); cultivation results from the TIS (blue bars). Data for 400, 800, and 1200 rpm in the SIS were averaged from three cultivations each. Dashed arrows indicate qualitative trends for each reactor setup.

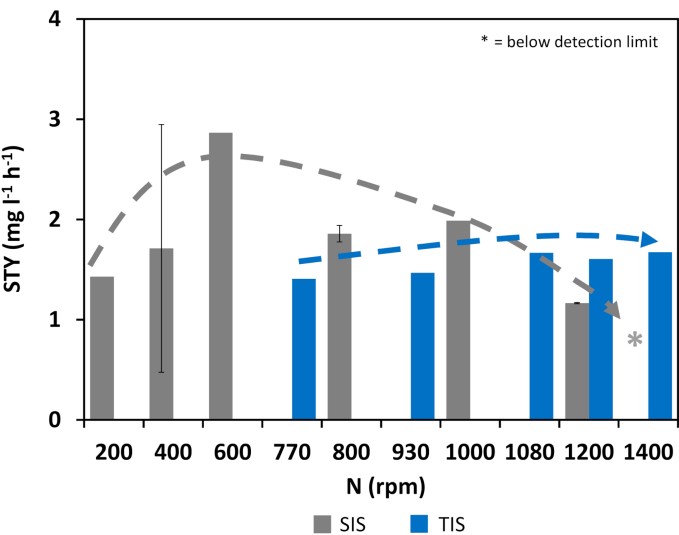

**Figure 6.** Antibody space time yields (STY) compared for experiments in both reactor setups: cultivation results from the SIS (grey bars); cultivation results from the TIS (blue bars). Data for 400, 800, and 1200 rpm in the SIS were averaged from three cultivations each. Dashed arrows indicate expected trends for each reactor setup.

### 3.3. Comparison of Cellular Behavior and Hydrodynamic Parameters

In the following, the presented experimental findings from cell cultures and hydrodynamic parameters are compared. In particular, it must be noted that the cells in the multi-stage reactors can be cultivated stably over a much broader operation range. This comparison will be made on the basis of various definitions for the energy dissipation rate including averaged and locally resolved ones. On one hand, parameters frequently used for process design, average power input P/V, and Kolmogorov length scale $\lambda$ based on average cell diameters $d_{c,av}$ are discussed. On the other hand, locally resolved parameters such as energy dissipation rate $\varepsilon$ distributions from CFD simulations, critical energy dissipation rate $\varepsilon_{krit}$ distributions calculated from cell size $d_c$ distributions, maximum energy dissipation rates $\varepsilon_{max}$, maximum energy dissipation rates from slice plots $\varepsilon_{max,SP}$ and variance coefficients of hydrodynamic parameters u, $\gamma$, and $\varepsilon$ will be considered.

The Kolmogorov length scale $\lambda$ [12,13] is a commonly used tool to evaluate the impact of hydrodynamics on mammalian cell cultures. With respect to flow induced cell damage, it is assumed that significant cell damage has to be expected when the size of the smallest turbulence eddy length is in the order of magnitude of the cell size. Therefore, for a known cell diameter $d_c$, the respective critical energy dissipation rate $\varepsilon_{krit}$ and hence, an appropriate agitation rate, can be estimated. Usually, the system averaged energy dissipation rate $\varepsilon_{av}$ is used to calculate the Kolmogorov length scale $\lambda$ and compared to a mean cell diameter $d_c$. The concept seems to work for microcarrier cultures, but is under discussion for suspension cells [14–19]

As an example, the average cell size $d_c$ of CHO DP-12 cells with about 10 µm would result in a critical energy dissipation rate $\varepsilon_{krit}$ of about 32 kW m$^{-3}$, which would be far above the calculated average energy dissipation rates $\varepsilon_{av}$ even at 1400 rpm, where a strong decrease in maximum viable cell density and antibody productivity were observed. Vice versa, an agitation rate calculated for process design with this approach would be way too high and might lead to undesired process behavior. Therefore, in order to evaluate a possible relationship between the cell size $d_c$ distribution and the Kolmogorov length scale $\lambda$, critical energy dissipation rate $\varepsilon_{krit}$ distributions (see Figures 7 and 8) were calculated from actual cell size $d_c$ distributions. Exemplary cell size $d_c$ distributions obtained from cultivation data are shown in the Appendix A (Figure A3). It was observed that cell size distributions neither expressively differed with changing agitation rates nor were dependent on the cultivation system.

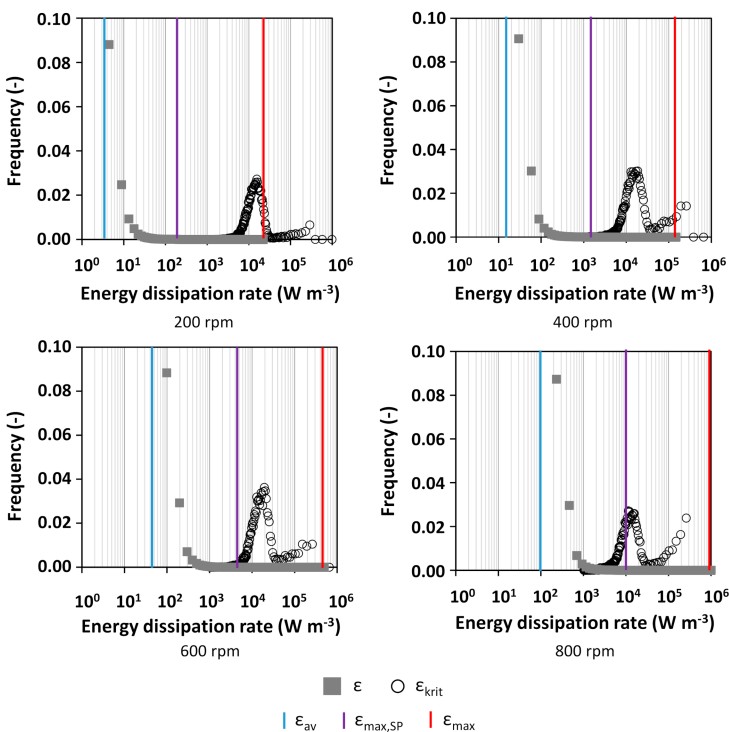

**Figure 7.** Critical energy dissipation rate $\varepsilon_{\text{krit}}$ distributions (black circles) calculated from the cell size distributions in the SIS compared to the energy dissipation rate distribution in the reactor system (grey squares) obtained from CFD data. The averaged energy dissipation rates from CFD simulations are indicated in blue, their maxima in red, maximum energy dissipation rates calculated from slice plots are indicated in purple.

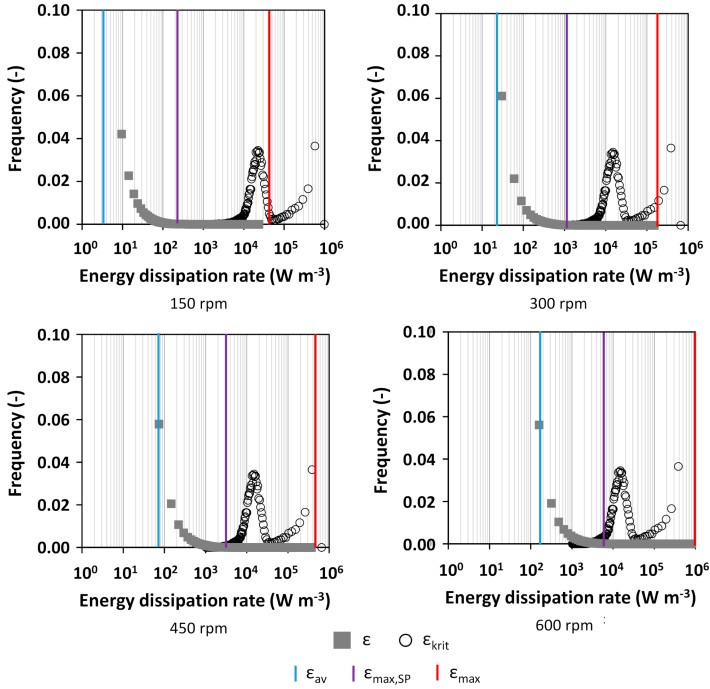

**Figure 8.** Critical energy dissipation rate distributions (black circles) in the TIS compared to the energy dissipation rate distribution in the reactor system (grey squares). The averaged energy dissipation rates from CFD simulations are indicated in blue, their maxima in red, maximum energy dissipation rates calculated from slice plots are indicated in purple.

The distribution of the normalized frequency of the energy dissipation rate $\varepsilon$ from CFD data for different agitation rates is shown in Figures 7 and 8. Furthermore, the averaged and maximum energy dissipation rates $\varepsilon_{av}$ and $\varepsilon_{max}$ from CFD simulations as well as the maxima of the slice plots are indicated in the diagrams.

In the cultivation experiments (see Section 3.2) with the SIS, no decrease in cell growth or antibody productivity was observed up to 800 rpm. The averaged energy dissipation rate $\varepsilon$ stayed a magnitude below the critical energy dissipation rate $\varepsilon_{krit}$ distribution curve, calculated from the cell size distributions. The maximum energy dissipation rates $\varepsilon_{max}$ from the CFD simulations were already above the main cell peak of the critical energy dissipation rate $\varepsilon_{krit}$ distribution at the lowest agitation rate. At 800 rpm, the maximum value was right below the peak that represents the main cell population. With increasing agitation rates, the maximum energy dissipation rate $\varepsilon_{max}$ also increases and, according to the Kolmogorov length scale hypothesis, for half of the cell population, shear related effects would have to be expected.

With CFD tools, it is possible to obtain normalized volumetric distributions of hydrodynamic parameters for the simulated systems [21,27]. Since the so far compared parameters might not provide satisfying information on the impact of shear forces, the actual normalized energy dissipation rate $\varepsilon$ distribution in the bioreactor was compared to the critical energy dissipation rate $\varepsilon_{krit}$ distribution, calculated from the cell size distributions (see Figures 7 and 8).

It was found that the energy dissipation rate in the largest part of the vessel volume stayed below the critical energy dissipation rate distribution curve. Only in a very small fraction did the energy dissipation rate in the reactor reach the area of the critical energy dissipation rate distribution. Although agitation rates above 800 rpm cannot be simulated, it can be concluded that the energy dissipation rate distribution curve of the reactor will move further into the critical energy dissipation rate distribution curve, according to extrapolations of the average and maximum values.

Additionally, for the TIS, the critical energy dissipation rate distributions were calculated and compared to the actual energy dissipation rate distribution in the cultivation system as well as the maximum energy dissipation rate from the slice plots. The results are shown in Figure 8.

For comparison of the different reactor systems, the plots for 600 rpm and 800 rpm in the SIS and 450 rpm in the TIS are shown in Figure 9. Their average volumetric power inputs were within the same order of magnitude (47 W m$^{-3}$ to 107 W m$^{-3}$) and clear differences are recognized.

In all cultivations in the TIS at 770 rpm to 1400 rpm, no noteworthy changes in cell growth or antibody productivity were present. From extrapolations of the slice plot maxima, which resulted in 64 kW m$^{-3}$ at 1400 rpm, which was just at the end of the main cell population peak (modal value at 10 kW m$^{-3}$ to 20 kW m$^{-3}$), it can be derived that the maximum from slice plots might not be a suitable parameter to make a connection between hydrodynamics and biology. Additionally, for this setup, the energy dissipation rate stayed below the critical energy dissipation rate distribution curve in the vast majority of the reactor volume. To have an optical impression of critical volume fractions, the areas in the cultivation system with energy dissipation rates above 1000 W m$^{-3}$ were identified. Detailed pictures can be found in the Appendix A (Figure A4).

For further analysis of both reactor systems, the hydrodynamic homogeneity was evaluated. Therefore, variance coefficients were calculated as the standard deviation from slice plot data divided by their mean value, which are shown in Figure 10.

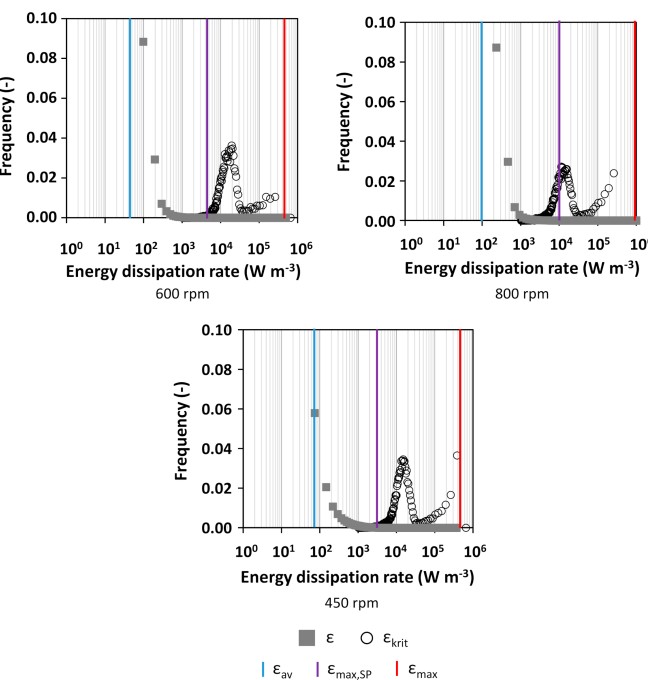

**Figure 9.** Critical energy dissipation rate distributions (black circles) compared to the energy dissipation rate distribution in the SIS (600 rpm and 800 rpm) and in the TIS (450 rpm) (grey squares). The averaged energy dissipation rates from CFD simulations are indicated in blue, their maxima in red. Maximum energy dissipation rates calculated from slice plots are indicated in purple.

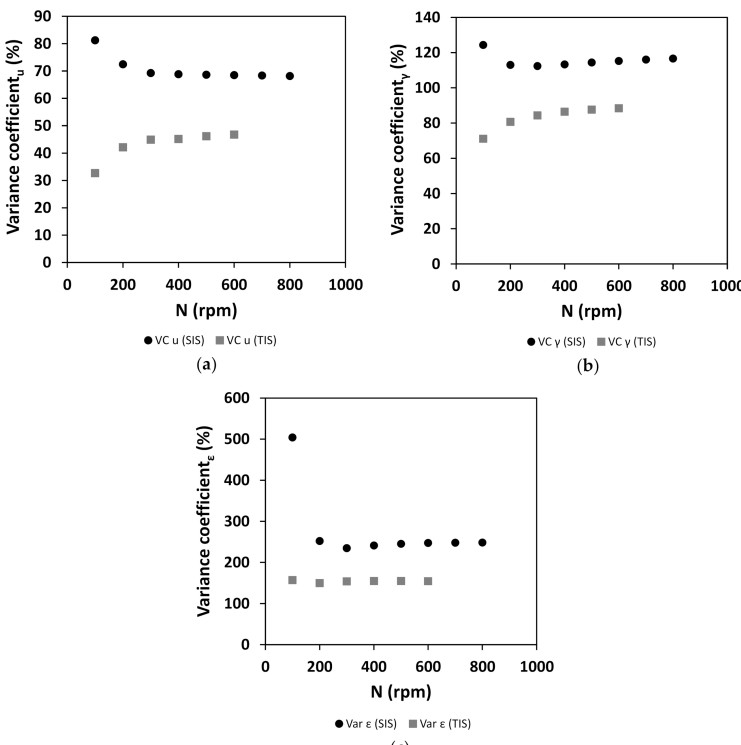

**Figure 10.** Variance coefficients for hydrodynamic parameters: (**a**) fluid velocity u, (**b**) shear rate $\gamma$, and (**c**) energy dissipation rate $\varepsilon$ in both cultivation systems. The coefficients were calculated from standard deviations from slice plot data divided by their mean value. They represent a measure of hydrodynamic homogeneity of the respective reactor system.

It can be seen that the variance coefficient of all three hydrodynamic parameters (fluid velocity u, shear rate $\gamma$, and energy dissipation rate $\varepsilon$) was higher in the SIS. This means that the SIS is a more heterogeneous system than the TIS. This might possibly be an explanation for the more stable process behavior of the TIS with respect to a changing agitation rate.

## 4. Discussion

The previously shown results from this study are discussed in the following under different aspects. The process stability that depends on hydrodynamics is debated and the application of the Kolmogorov length scale hypothesis with modern CFD methods is highlighted. Finally, consequences for the scale-up and scale-down of the bioreactors are drawn.

### 4.1. Hydrodynamic Differences for Single and Multi-Stage Impellers

Certain differences in flow pattern, velocity, and other hydrodynamic parameters have been observed for multiple impellers in the same stirred reactor system. It was found from CFD studies that the system averaged fluid velocity, shear rate, and energy dissipation increased with the addition of multiple impellers. The maximum shear rate and energy dissipation rate also increased in the reactor setup with multiple impellers. Only the maximum fluid velocity remained almost constant as it depends on the agitation rate solely. Overall, it can be concluded that the use of multiple impellers leads to a more uniform fluid flow and therefore to a lower variance coefficient of the hydrodynamic parameters.

### 4.2. Stability of Cell Culture Processes Related to Locally Resolved Hydrodynamics

Cultivation results for both reactor setups showed clear differences in the process behavior with respect to changing agitation rates. While the maximum cell density decreased up to 60% in the SIS, and antibody productivity decreased up to 100% above 1000 rpm (averaged 203 W m$^{-3}$), no changes were recorded for agitation rates up to 1400 rpm (averaged 4742 W m$^{-3}$) in the TIS. Platas et al. [28] reported a rather stable process behavior at higher agitation rates when using a bioreactor with multiple impellers instead of a single impeller. We identified that local hydrodynamic maxima and hydrodynamic heterogeneities, represented by variant coefficients were less pronounced in the TIS at the same averaged energy dissipation rate, resulting in a more homogeneous fluid flow. From the cultivation data, it could be concluded that this leads to a more robust process behavior.

### 4.3. Evaluation of Concepts for Estimation of Shear Related Parameters

From the results of the cultivation experiments, it can be concluded that the average volumetric power input or energy dissipation rate is clearly not suitable as a universal shear-related parameter for bioprocess design. The two examined reactor systems showed a different process behavior for the same averaged energy dissipation rates. Even if the average dissipation rate increases way faster in the TIS with increasing agitation rate, the process is still more stable with respect to the agitation rate in the TIS. Maximum energy dissipation rates are also not suitable.

None of the parameters calculated from CFD data corresponded to the cultivation process behavior with changing agitation rate for both cultivation systems. Hence, it can be assumed that these parameters are not sufficient as indicators for the impact of hydrodynamics on the cells. However, solely, the maximum energy dissipation rates from the slice plots $\varepsilon_{\mathrm{max,SP}}$ could possibly show a connection to the cellular effects.

The Kolmogorov length scale is commonly used to describe the size of smallest eddies that might lead to cell damage [14–19]. In this study, it was calculated in the form of distributions instead of one fixed length for the whole bioreactor due to strong hydrodynamic heterogeneities in the cultivation systems. In addition, cell size distributions instead of a mean cell size were considered. The resulting critical energy dissipation rate distribution, calculated from cell size distributions, was compared to the averaged and

maximum energy dissipation rate as well as to the maxima from slice plots and volumetric distributions of the energy dissipation rate, all from CFD simulations.

It was found that the Kolmogorov length scale is not suitable to describe a link between hydrodynamics and cell damage, if the averaged or maximum energy dissipation rate is used since they do not correlate with any biological observations (Figures 3–6). Furthermore, critical volume fractions of the cultivation vessels do not deliver the desired insights, despite the fact that the energy dissipation rate distribution curve moves into the critical region with increasing agitation rate. Nevertheless, extrapolations of the maximum energy dissipation rates from slice plots showed a slightly faster increase for the TIS. Since cells grew with roughly the same growth rate up to the highest agitation rate in this setup, this parameter does not seem to be suitable to describe a link between cell growth and hydrodynamics, but still seems to be a closer estimation than the other parameters.

The hydrodynamic heterogeneity of a bioreactor, which was quantified in this study with variance coefficients of the hydrodynamic parameters fluid velocity, shear rate, and energy dissipation rate, might be a useful parameter to estimate the suitability of a cultivation system. The calculated variance coefficients of all hydrodynamic parameters were higher in the TIS than in the SIS, which might explain the rather stable process behavior in multiple impeller systems due to the improved hydrodynamic homogeneity.

### 4.4. Learning for Reactor Scale-Up and Scale-Down

When performing scale-up of bioreactors, common criteria that are kept constant are the impeller tip speed or the averaged power input [7–10]. In this study, the averaged power input increased much faster with increasing agitation rate in the TIS than in the SIS. However, no prominent differences in process behavior were found in cultivations at different agitation rates in the TIS, while in the SIS, the maximum cell densities and antibody productivities were reduced at high agitation rates. Consequently, the cultivation data showed that the biological process behavior does not solely depend on single process parameters such as the averaged power input, which are commonly used in process design, but rather on their homogeneity. Since maxima of hydrodynamic parameters also increased faster in the TIS and no relevant changes in process behavior were recorded, it is likely that the whole reactor system and locally resolved hydrodynamics need to be considered. Therefore, locally resolved hydrodynamic parameters were carved out of the CFD data as slice-plots and as volume fraction specific distributions. It was observed that strong hydrodynamic heterogeneities were present in stirred tank reactor systems, which became steeper with increasing agitation rate. A multi-parametrical approach considering locally resolved hydrodynamics resulting from geometrical characteristics could be the consequence for process scale-up. In turn, the investigation of local hydrodynamic effects has been the focus of late. Local distributions of the shear stress and the Kolmogorov length scale have been evaluated for spinner flasks, but have not been compared to cell size distributions thus far [27]. Furthermore, the calculation of cellular residence times in differently mixed reactor areas could provide deeper insights on the actual impact of hydrodynamic phenomena on cellular behavior [22,29]. Another recent study showed that it is worth considering the local distributions of conventional process design parameters such as $k_L a$ [10]. In addition, the local determination of the power input shows promising results for the design of cultivation processes for human mesenchymal stem cells. These approaches could be combined with novel uncertainty-based model evaluation methods [6,30]. It is noteworthy that the six blade impellers, specifically designed for the TIS in this study, worked immediately for the cultivation of CHO DP-12 cells.

### 5. Conclusions

In this study, parameters obtained from CFD simulations were linked to cell culture cultivation experiments to investigate the influence of hydrodynamic indifferent reactor setups on cell growth and antibody productivity. It was shown that hydrodynamically more uniform conditions and the resulting flatter hydrodynamic profiles might be the

reason for the broader stable operational space regarding the agitation rate of a stirred tank with multiple impellers, which was previously observed by Platas et al. [28]. The evaluation of the CFD results clearly showed stronger pronounced hydrodynamic heterogeneities for the same power input in single impeller setups. Hence, the use of conventional, mostly averaged process design parameters, needs to be questioned and rather, local gradients should be considered, especially for scale-up [22,29]. Furthermore, it was found that the Kolmogorov length scale hypothesis might not be appropriate to describe the influence of hydrodynamics on mammalian cell culture processes without a hydrodynamic characterization of the full cultivation system including all hydrodynamic gradients. Overall, a deeper insight into local hydrodynamic gradients in cell culture reactors was obtained, but the reliable prediction of design parameters prior to an experimental evaluation still remains difficult.

**Supplementary Materials:** The following supporting information can be downloaded at: https://www.mdpi.com/article/10.3390/pr10010107/s1, File S1: Supplementary File SIS Model Report.pdf, File S2: Supplementary File TIS Model Report.pdf.

**Author Contributions:** Conceptualization, F.F., J.M., R.P. and M.S.; Methodology, F.F.; Software, F.F. and J.B.; Validation, F.F. and J.M.; Formal analysis, F.F. and E.A.; Investigation, F.F., E.A. and J.B.; Resources, R.P.; Data curation, F.F., E.A. and J.B.; Writing—original draft preparation, F.F., J.M., and R.P.; Writing—review and editing, M.S.; Project administration, J.M. and R.P. All authors have read and agreed to the published version of the manuscript.

**Funding:** This research received no external funding.

**Institutional Review Board Statement:** Not applicable.

**Informed Consent Statement:** Not applicable.

**Conflicts of Interest:** The authors declare no conflict of interest.

## Appendix A

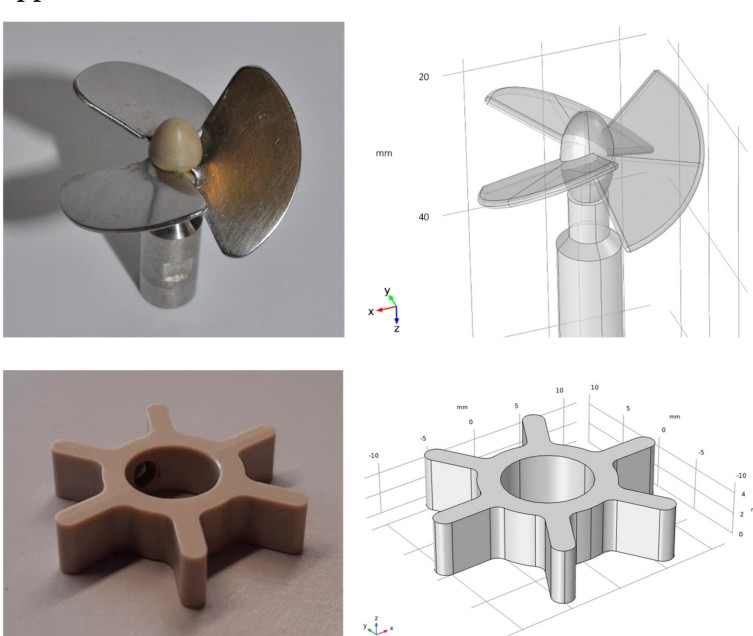

**Figure A1.** Pitched blade impeller (**above**) and six blade impeller (**below**) with corresponding CAD models. For the pitched blade impeller, a Newton number Ne of 0.35 was determined with an empiric correlation [7], while for the six blade impeller, 3.95 was considered, following another empiric correlation [31].

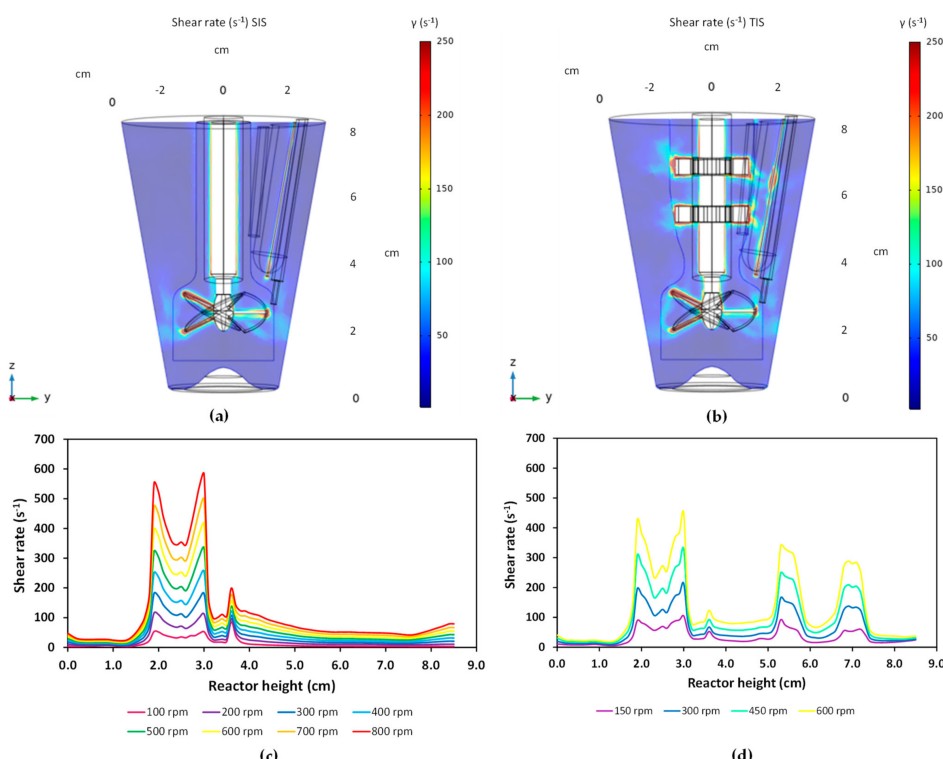

**Figure A2.** Shear rate $\gamma$ in the SIS and TIS: (**a**) shear rate in the SIS; (**b**) shear rate in the TIS. The shown data correspond to an agitation rate of 400 rpm. (**d**) Mean shear rate along the reactor height (slice plot) in the SIS for different agitation rates; (**d**) mean shear rate along the reactor height (slice plot) in the TIS for different agitation rates. In (**c**,**d**), the shear rate was averaged across the diameter in steps of one millimeter along the reactor height.

In both cases, for most parts of the reactor systems, a quite uniform distribution of the shear rate could be observed. The mean shear rate did not exceed $100\ \mathrm{s}^{-1}$, but reached almost $600\ \mathrm{s}^{-1}$ at the height of the stirrer in the SIS, and the shear rate showed values of up to $9167\ \mathrm{s}^{-1}$ at 400 rpm. Only small parts of the reactor reached values above $1000\ \mathrm{s}^{-1}$, namely at the stirrer. In the largest part along the reactor height, the mean shear rate did not exceed $100\ \mathrm{s}^{-1}$, but reached almost $600\ \mathrm{s}^{-1}$ at the height of the stirrer. The maximum values in the TIS were similar for equal agitation rates with $9383\ \mathrm{s}^{-1}$ at 400 rpm, but again, two additional peaks could be found in the slice plots, representing the added impellers. However, the values between the stirrers were still comparably low, below $100\ \mathrm{s}^{-1}$.

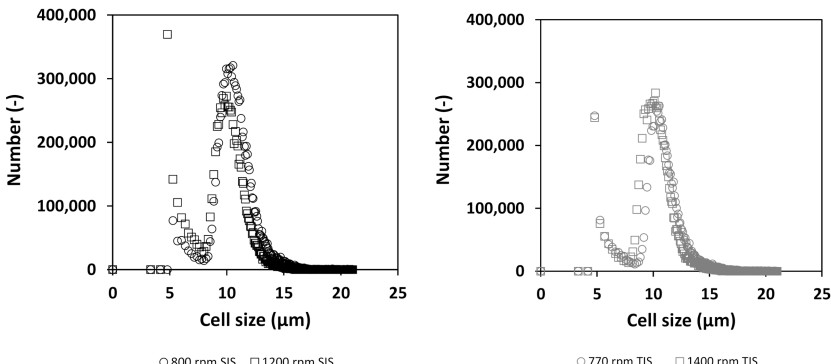

**Figure A3.** Exemplary cell size distributions from cultivation data in the SIS (800 rpm and 1200 rpm) and in the TIS (770 rpm and 1400 rpm). Distributions for all cultivations were taken from samples at 72 h cultivation time.

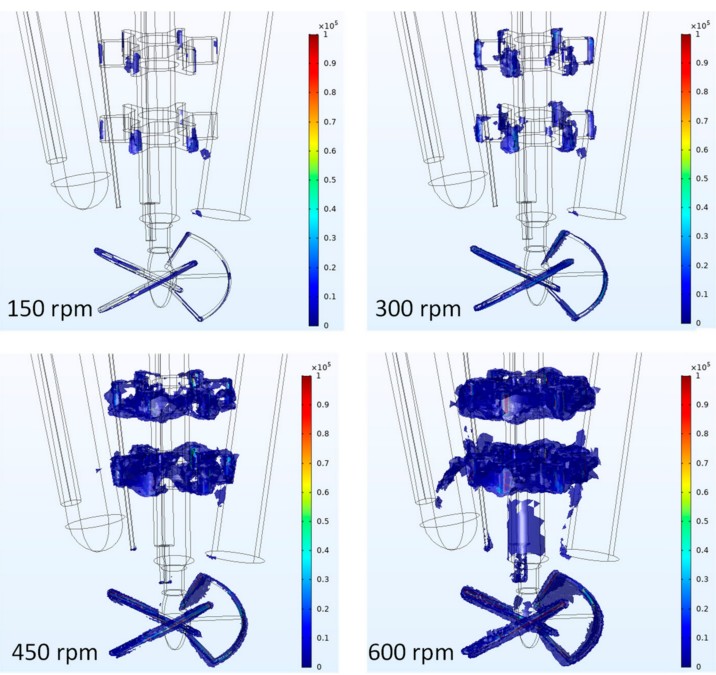

**Figure A4.** Areas in the TIS with energy dissipation rates above 1000 W m$^{-3}$.

The regions above 1000 W m$^{-3}$ are all located in thin layers on the pitched blade impellers, small areas around the six blade impellers, and very small parts at the edges of probes and other inserts.

With a defined threshold for the critical energy dissipation rate, the critical volume fractions of the cultivation system can be calculated from CFD data. They are linearly dependent on the power input or energy dissipation rate. Due to the respective number of impellers, the different reactor setups result in different critical reactor volume fractions for the same power input. The critical areas were smaller for equal averaged power inputs in the TIS due to flatter gradients. Cultivation experiments in the SIS showed a decrease in cell growth and antibody productivity above 1000 rpm. In the TIS, no changes were recorded. Since the critical reactor volume for 1400 rpm was about five-fold higher in the TIS (0.45% at 4.7 kW m$^{-3}$) than in the SIS (0.07% at 465 W m$^{-3}$), it can be concluded that the critical volume fraction might not be a suitable parameter to also evaluate the influence of hydrodynamics on the process.

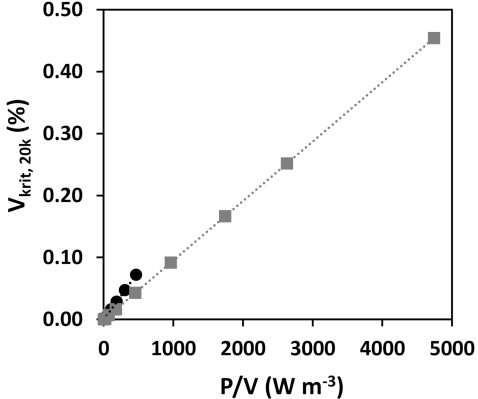

**Figure A5.** Critical reactor volume fractions with energy dissipation rates above 20 kW m$^{-3}$ plotted over the corresponding averaged power input. Black circles represent values for the SIS, grey squares those of the TIS.

**Table A1.** Hydrodynamic parameters in the SIS based on CFD simulations. All values above 800 rpm were extrapolated linear (velocity and shear rate) or quadratic, respectively (energy dissipation rate).

| N (rpm) | $u_{av}$ (m s$^{-1}$) | $\gamma_{av}$ (s$^{-1}$) | $\varepsilon_{av}$ (W m$^{-3}$) | $\varepsilon_{max}$ (W m$^{-3}$) | $\varepsilon_{max,SP}$ (W m$^{-3}$) |
|---|---|---|---|---|---|
| 200 | 0.032 | 10.06 | 2.39 | 22,169 | 214 |
| 400 | 0.068 | 21.23 | 15.35 | 146,510 | 1575 |
| 600 | 0.103 | 32.22 | 47.35 | 491,767 | 4959 |
| 800 | 0.138 | 43.15 | 107.12 | 1,159,628 | 11,001 |
| 1000 | 0.171 | 53.56 | 203.56 | 2,265,066 | 20,239 |
| 1200 | 0.205 | 64.27 | 345.05 | 3,932,425 | 33,194 |
| 1400 | 0.240 | 74.99 | 540.50 | 6,286,050 | 50,385 |

**Table A2.** Hydrodynamic parameters in the TIS based on CFD simulations. All values above 600 rpm were extrapolated linearly (velocity and shear rate) or quadratically, respectively (energy dissipation rate).

| N (rpm) | $u_{av}$ (m s$^{-1}$) | $\gamma_{av}$ (s$^{-1}$) | $\varepsilon_{av}$ (W m$^{-3}$) | $\varepsilon_{max}$ (W m$^{-3}$) | $\varepsilon_{max,SP}$ (W m$^{-3}$) |
|---|---|---|---|---|---|
| 150 | 0.039 | 13.24 | 3.63 | 45,738 | 221 |
| 300 | 0.081 | 28.62 | 23.86 | 194,000 | 1118 |
| 450 | 0.125 | 43.93 | 72.90 | 479,000 | 3014 |
| 600 | 0.159 | 55.53 | 176.93 | 1,010,000 | 6247 |
| 770 | 0.208 | 72.98 | 454.35 | 2,007,732 | 12,103 |
| 930 | 0.252 | 88.15 | 964.25 | 3,401,191 | 20,258 |
| 1080 | 0.292 | 102.36 | 1744.79 | 5,172,104 | 30,686 |
| 1200 | 0.325 | 113.74 | 2633.29 | 6,952,684 | 41,252 |
| 1400 | 0.379 | 132.69 | 4742.49 | 10,725,752 | 63,867 |

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
