# Peer review of "New Insights from Locally Resolved Hydrodynamics in Stirred Cell Culture Reactors"

_processes, doi:10.3390/pr10010107_

Round 1

Reviewer 1 Report

I read with interest the manuscript entitled "New Insights from Locally Resolved Hydrodynamics in Stirred Cell Culture Reactors". In this work, the authors explored the link between local hydrodynamics and biological process behavior in two different stirred tank bioreactor configurations. The hydrodynamics developing locally within the bioreactors was investigated by CFD modeling and cultivation tests were performed for assessing the biological responses of the process with respect to the reactor setup, finding that more uniform flow patterns, obtained by the use of multiple impellers, lead to a more stable process behavior at increasing agitation rates.

The manuscript deals with an interesting topic in view of optimization and standardization of scalable manufacturing processes, however in order to make the manuscript suitable for publication it should be refined and some parts should be clarified and discussed more in detail, for these reasons a revision is required. In the following, you can find the detailed comments from the reviewer.

Major comments

 Introduction

The authors should provide more information about the application of the CFD approach for supporting the design and the operation of bioreactors. Please check the following references:

Jossen V, Schirmer C, Mostafa Sindi D, et al. Theoretical and Practical Issues That Are Relevant When Scaling Up hMSC Microcarrier Production Processes. Stem Cells Int. 2016;2016:4760414. doi:10.1155/2016/4760414

Möller J, Pörtner R. Digital Twins for Tissue Culture Techniques—Concepts, Expectations, and State of the Art. Processes. 2021; 9(3):447. https://doi.org/10.3390/pr9030447

Stephan Kaiser, Valentin Jossen, Carmen Schirmaier, Prof. Dieter Eibl, Dr. Silke Brill, Christian van den Bos, Regine Eibl. Fluid Flow and Cell Proliferation of Mesenchymal Adipose-Derived Stem Cells in Small-Scale, Stirred, Single-Use Bioreactors.       10.1002/cite.201200180

Isu G, Morbiducci U, De Nisco G, et al. Modeling methodology for defining a priori the hydrodynamics of a dynamic suspension bioreactor. Application to human induced pluripotent stem cell culture. J Biomech. 2019;94:99-106. doi:10.1016/j.jbiomech.2019.07.021

Materials and methods

At lines 111-2 the authors state that “Reactor vessel and impellers were modelled as accurate as possible and necessary”. Authors should describe which elements of the internal geometry of the bioreactors were considered and which elements were neglected and the reason behind these choices.

At line 116 the authors state that: “The geometry was meshed to roughly 1.5 to 2.5 million mesh elements”. Are the two numbers respectively referred to the SIS and the TIS? If so, authors should specify the respective approximate number of elements for each geometry.

At lines 117-9 the authors state that:” Due to numerical reasons agitation rates of up to 800 rpm for the reactor setup with single impeller and 600 rpm for the reactor setup with three impellers were simulated”. Then, to compare the CFD results with experimental results performed with higher agitation rates, the authors computed the hydrodynamic parameters up to 1400 rpm, based on extrapolations. This technique is briefly mentioned in the Results section (lines 201-203) and detailed only in Table 1 caption (line 518) and Table 2 caption (line 521). The use of this extrapolation technique should be described in the Methods section.

At line 96, the authors state: “cultivation experiments were performed at different agitation rates”. This information should be clarified providing the list of tested agitation rates.

Results

Lines 139-143 explain how the mean fluid velocity and mean energy dissipation rate, displayed in Figures 1c-d and 2c-d, were computed. The step size for the averaging is specified in Figure 1 caption (line 159-160) and Figure 2 caption (line 172-173). For clarity reasons, this information should be reported in the main text.

At lines 201-203 the authors state that: “Further correlations for agitation rates, averaged dissipation rates εav and maximum energy dissipation rates εmax for both systems can be found in Table 1 and 2 in the appendix.” It should be mentioned (referring to the methods section) how these parameters were computed.

At line 234, the authors refer for the first time to the “space time yield (STY)”, this parameter should be introduced and clarified in the Methods section.

Section 3.3 “Comparison of Cellular Behavior and Hydrodynamic Parameters“, introduces two parameters used to compare the flow developed inside the bioreactor for the two different setups. In detail, at lines 259-69 the authors provide the definition of Kolmogorov length scale and describe the rationale for using this parameter, while at lines 275-80 they introduce the concept of critical energy dissipation rate. Since computing these parameters is a method applied for the evaluation of the Results of the CFD analysis, the description and application of these two parameters should be described in the Methods section.

At lines 293-295 the authors state that: “Hence, it can be assumed that these parameters are not sufficient indicators for the impact of hydrodynamics on the cells. Solely, the maximum energy dissipation rates from the slice plots εmax,SP could possibly indicate a connection”. This paragraph should be moved to the Discussion.

At lines 339-342 the authors state that: “From extrapolations of the slice plot maxima, which result in 38 kW m-3 at 1400 rpm, which is just at the end of the main cell population peak (median at 10 kW m-3 to 20 kW m-3), it can be derived that the maximum from slice plots might not be a suitable parameter to make a connection between hydrodynamics and biology.”. However, the reported result (38 kW m-3) is not consistent with the ε max,SP value reported in Table 2 for the same condition, which is 63867 W m-3. The authors should check and in case modify the text.

Minor comments

Lines 138-139 should better specify what is depicted in Figure 1 and 2 panels, explaining which panels refer to the SIS setup and which to the TIS setup.

The authors should refer to the “single- impeller setup (SIS) and the three-impeller setup (TIS)” in Methods section (lines 97-8) instead of in Results section (lines 123-4).

At Line 165-167, the authors use the term “homogenized”, while the term “uniform” would be more appropriate.

In Figure 5 and 6, the authors should add a legend about the two adopted colors.

Author Response

Dear Reviewers,

We thank you very much for reviewing our manuscript. We kindly acknowledge that we received comments with such an in-depth understanding of the related research field, which helped us to strengthen the outline of the manuscript. Please find a list of the questions and suggestions and our corresponding answers below.

Kind regards,

Fabian Freiberger, Jens Budde, Eda AteÅŸ, Michael Schlüter, Ralf Pörtner, and Johannes Möller

Reviewer 2 Report

New Insights from Locally Resolved Hydrodynamics in Stirred 3 Cell Culture Reactors

Abstract

  1. The abstract should be more well written with specific findings rather than generalizations.

Introduction

  1. Avoid using e.g. in journal writing.
  2. The achievement of previous research is not clearly introduced.
  3. The introduction did not clearly introduce the common flow rate, shear force, flow velocity, shear rate and power input or hydrodynamic rate of previous studies.
  4. Do clearly state the objective of the current work.

Content

  1. “Only at the end of the cultivations at cell densities of around 10 × 106 cells ml-1”. The cell density at 10 x 10^6 cells/ml is low for a reactor. Please check carefully. Cell density of post-culture is lower than the pre-culture. Is this correct? There is also grammatical error in this sentence.

  1. “Samples above 2 × 106 cells ml 104 -1 were diluted ten folds before staining”. -The exact cell density should be determined before staining and not the the pre-dilution cell density. This procedure is awkward and not giving the clear information.

  1. The constitutive model for CFD study using Multiphysics comsol is not clearly explained and the model was not included. The parameters or properties set for the CFD model is unavailable.
  2. The effects of the flow pattern of the CFD model to the cell growth was not discussed and analyzed. The results were not compared with previous work also.

  1. Legend of Figure 4 should write viability in percentage and not just viability.

  1. The discussion should compare with previous work.

Conclusion

  1. The conclusion should explicitly state which parameters provide optimum cell growth and antibody production, and not just generalization. The conclusion are not supported by results.

References

  1. Please remove self-citations

Author Response

(The authors gave the same response as above.)

Reviewer 3 Report

This paper evaluates hydrodynamic behavior of stirred cell culture reactors, and compares the performances between two impeller types via both CFD and experimental operation.

The structure of this paper is in good organization, so as the enriched contents introduced in this study. However, some suggestions need to be considered as well,

  1. Line 23, the last sentence adds a certain degree of unclearness in the abstract. Try to avoid this description by replacing the “newly gained insights” “deeper understanding” “process” with more concrete vision.
  2. Keywords fail to reveal the true value of this work, they are all way too “big” and common to attach the topic of this study. you can add one keyword to represent the feature close to the study.
  3. Line 92 Give full name for the first appeared specific letters “DO”
  4. Line 94 Wrong chemical expression as “NaCO3.
  5. Section 2.3: This section needs more detailed introduction. As CFD simulation you used as a tool for revealing the flow field of the reactor. You should give more information for the model you used (steady/ unsteady state? Species transport model? Whether or not the heat transfer or cavitation is involved in practical process? What are the differences of the settings between current and cases database in comsol? How do you verify its accuracy? … this section needs to be enriched.
  6. line 117: what are the numerical reasons for setting the unmatched rotation speed? (Many cases with different agitation rates were compared, but only 600 rpm was comparable). If frequent divergence occurs in that operation range, the accuracy of the model or settings must be questioned.
  7. line 141, if you mean the absolute value of the velocity, better explains as” mean velocity magnitude”
  8. Section 3.2: what is the reason behind the different VCD distributions of STY and TIS? From the presented results in this section, is seems that the TIS achieves similar performances only at higher agitation rates compared with STY which could consume less energy in slow rotation?
  9. Fig 7- Fig 9: add legends for better readability.
  10. As for the conclusion part, this part needs to be strengthen so that readers can quickly catch important findings of this study. As some concrete findings are introduced in discussion, this part can add more vision for the future work, and your prospect for the uncertain correlations failed to quantitatively revealed? The whole work of this study is interesting and of scientific values, but the conclusion is too hazy and weak to represent your work.

Author Response

(The authors gave the same response as above.)
